# Viscosity Modification of Polymerizable Bicontinuous Microemulsion by Controlled Radical Polymerization for Membrane Coating Applications

**DOI:** 10.3390/membranes10090246

**Published:** 2020-09-21

**Authors:** Ephraim Gukelberger, Christian Hitzel, Raffaella Mancuso, Francesco Galiano, Mauro Daniel Luigi Bruno, Roberto Simonutti, Bartolo Gabriele, Alberto Figoli, Jan Hoinkis

**Affiliations:** 1Laboratory of Industrial and Synthetic Organic Chemistry (LISOC), Department of Chemistry and Chemical Technologies, University of Calabria, 87036 Rende (CS), Italy; ephraim.gukelberger@gmail.com (E.G.); raffaella.mancuso@unical.it (R.M.); bartolo.gabriele@unical.it (B.G.); 2Center of Applied Research (CAR), Karlsruhe University of Applied Sciences, 76133 Karlsruhe, Germany; christianhitzel@web.de; 3Institute on Membrane Technology, National Research Council (ITM-CNR), 87036 Rende (CS), Italy; f.galiano@itm.cnr.it (F.G.); a.figoli@itm.cnr.it (A.F.); 4Department of Physics, University of Calabria, 87036 Rende (CS), Italy; bruno.mauro89@gmail.com; 5Department of Materials Science, University of Milan-Bicocca, 20126 Milan, Italy; roberto.simonutti@unimib.it

**Keywords:** viscosity modification, polymerizable bicontinuous microemulsion, controlled radical polymerization, membrane coating, wastewater treatment

## Abstract

Membrane modification is becoming ever more relevant for mitigating fouling phenomena within wastewater treatment applications. Past research included a novel low-fouling coating using polymerizable bicontinuous microemulsion (PBM) induced by UV-LED polymerization. This additional cover layer deteriorated the filtration capacity significantly, potentially due to the observed high pore intrusion of the liquid PBM prior to the casting process. Therefore, this work addressed an innovative experimental protocol for controlling the viscosity of polymerizable bicontinuous microemulsions (PBM) before casting on commercial ultrafiltration (UF) membranes. Prior to the coating procedure, the PBM viscosity modulation was carried out by controlled radical polymerization (CRP). The regulation was conducted by introducing the radical inhibitor 2,2,6,6-tetramethylpiperidine 1-oxyl after a certain time (CRP time). The ensuing controlled radical polymerized PBM (CRP-PBM) showed a higher viscosity than the original unpolymerized PBM, as confirmed by rheological measurements. Nevertheless, the resulting CRP-PBM-cast membranes had a lower permeability in water filtration experiments despite a higher viscosity and potentially lower pore intrusion. This result is due to different polymeric structures of the differently polymerized PBM, as confirmed by solid-state nuclear magnetic resonance (NMR) investigations. The findings can be useful for future developments in the membrane science field for production of specific membrane-coating layers for diverse applications.

## 1. Introduction

Polymerizable bicontinuous microemulsion (PBM) polymerized with redox initiation has demonstrated its potential as a low-fouling membrane coating for wastewater treatment application [1,2]. The first theoretical and practical experiments about the micelle formation and polymerization of microemulsions for the production of nanostructured membranes were considered by Figoli [3]. Galiano et al. [1,4] further investigated fundamental coherences between the employed materials such as the conductivity and micelle formulation using the titration method.

They developed the microemulsion composition leading to a polymerized PBM coating with the best possible low-fouling characteristics. For instance, the home-made surfactant Acryloyloxyundecyl-triethyl ammonium bromide contains a quaternary ammonium salt group, which is highly antimicrobial especially against gram-negative bacterium and E.Coli [1]. In conjunction with the co-surfactant 2-hydroxylethylmeth-acrylate^,^ the coating adds low-fouling properties to the pristine, commercial membranes.

As a result of the study, Deowan et al. [2] applied this novel coating to porous, ultrafiltration (UF) polyethersulfone (PES) membranes within a lab-scale submerged membrane bioreactor (MBR). They verified the validity of the approach in the efficient treatment of wastewater in the textile industry. In the framework of the European funded BioNexGen project [5], the redox-polymerized PBM coating was tested at pilot scale using a 0.33 m^2^ coated membrane surface. The pilot scale trials demonstrated a higher performance of the coated membrane than its commercial counterpart during long-term operation close to the critical flux.

An alternative, based on photo-initiated UV-LED polymerization of PBM, was investigated by Schmidt [6] and Galiano et al. [7]. In this case, the coating curing times were successfully reduced down to 30 s. Nevertheless, PBM polymerization induced by UV-LED light led to a lower filtration capacity in the model foulant tests inside a cross-flow unit treating a 100 mg·L^−1^ humic acid solution (pH 9) [7], probably due to a higher intrusion into the membrane pores with respect to redox-polymerized PBM.

In order to achieve comparable results in terms of permeability with respect to the redox-based polymerization, the investigation presented in this work was aimed at achieving a higher viscosity of the PBM prior to membrane casting and UV-LED curing in order to effectively reduce the PBM intrusion into the membrane capillaries. To achieve this goal, the PMB was subjected to a controlled radical polymerization stage before being cast on a commercial UF PES membrane surface and subjected to UV-LED curing.

Controlled radical polymerization (CRP) can be achieved via reaction retarding or complete stopping and as a consequence results in a modification of the molecular weights and polydispersity index (PDI) of the final polymer. For instance, Platkowski and Reichert [8] showed a way to retard and stop homogeneous solution polymerization reactions of polyacrylic acid using 4-hydroxy-2,2,6,6-tetramethylpiperidinyloxyl. They also developed a mathematical model to calculate the polymerization propagation after adding different amounts of an inhibitor with a high impact on polymer chain growth. Nicolas et al. [9] reviewed and discussed fundamental aspects of the nitroxide-mediated polymerization (NMP) which enabled the synthesis of well-defined and complex macromolecular architectures. Menter [10] introduced a practical way of tailoring the molecular chain lengths and thereby influencing the final product specifications of polyacrylamide gels by using different amounts of photo-initiators. It was found that higher initiator amounts led to shorter molecular chain lengths, thus resulting in a higher polymer crosslink density. In terms of the low-fouling PBM membrane coating, it would also make sense to change the coating layer texture and pore structure with suitable modulation of the polymerization propagation.

This work describes the approach developed for PBM viscosity modification by a PBM radical polymerization control in detail. Rheological measurements were carried out for the thus modified PBM samples (controlled radical polymerized PBMs, CRP-PBMs), obtained by varying the controlled radical polymerization times. Coating layers were then obtained by UV-LED curing of CRP-PBMs with different viscosities, after casting them on the surface of commercial UF PES membranes. The permeability of the surface functionalized membranes was tested with a membrane cross-flow unit and a 0.0085 m^2^ coated membrane area. Using the obtained data, the relationship between the polymerization time, the dynamic viscosity of the CRP-PBM and the water permeability was established. In order to better understand the differences in morphology of the PBM mixtures, differently polymerized samples were examined utilizing ^13^C magic angle spinning nuclear magnetic resonance (MAS-NMR) spectroscopy.

## 2. Materials and Methods

### 2.1. Chemicals Used

Table 1 lists all the chemicals used in this work: the basic PBM components (1); the photo-initiator (2) activated by UV-LED light; the chemicals used for redox initiation (3); and the strong radical inhibitor 2,2,6,6-tetramethylpiperidine 1-oxyl (TEMPO) (4) for stopping/retarding the polymerization reaction. All chemicals, except acryloyloxyundecyltriethyl ammonium bromide (AUTEAB) surfactant and DI water, were purchased from *Sigma-Aldrich* (St. Louis, MO, United States) with the given purities shown in Table 1. AUTEAB is a lab-made surfactant synthesized as described previously [1]. Ultrapure DI water was produced with a Millipore Direct-Q3 purification system (*Merck*, Darmstadt, Germany).

### 2.2. Preparation of Polymerizable Bicontinuous Microemulsions (PBM)

The method for the PBM preparation reported by Galiano et al. [1] was used with the mass fraction of each component shown in Table 1. The antimicrobial [11] and polymerizable AUTEAB surfactant was chemically bound to the solid PBM matrix upon co-polymerization. Other components used were the cross-linking agent ethylene glycol dimethacrylate (EGDMA) and the co-surfactant 2-hydroxyethylmethacrylate (HEMA). All components were mixed successively at room temperature using a magnetic stirrer to obtain a bicontinuous microemulsion, where the two phases coexist in interconnected domains [2].

### 2.3. Pore Intrusion Potential for UF PES Membranes

The pore intrusion of the PBM-coated membrane, characterized by the migration of the liquid PBM solution into the porous PES layer, was first observed by preliminary visual trials. The quantification of the pore intrusion depth was then established using confocal light microscopy. To this end, fluorescent SiO_2_ nanoparticles were added to the liquid PBM prior to the coating process. The nanoparticles were synthesized by immobilizing tris(phenantroline)ruthenium(II) chloride [RuCl_2_(phen)_3_] in silica nanoparticles prepared according to the Stöber method [12]. 30 mg of nanoparticles were dispersed within a 1 mL MMA pure phase. Sonication twice for 15 min prevented agglomeration and enabled a homogeneous distribution within the test volume for the best optical results. In the course of the coating process, the nanoparticles flowed partly inside the pores and were entrapped within the liquid-solid phase change. Subsequent to the coating procedure, the membrane was cut in cryogenic, liquid nitrogen for cross-sectional investigation with a confocal microscope *TCS SP8* (*Leica*). An examination of the pore intrusion was executed with an excitement wavelength of 561 nm at a 552–625 nm acquisition window. A 20 x magnification range was selected and the samples were immersed in DI-water to control refraction of the light path.

### 2.4. PBM Controlled Radical Polymerization (CRP) Set-up

PBM controlled radical polymerization (CRP) was carried out by redox initiation (with ammonium perfulfate (APS), in the presence of *N*,*N*,*N*′,*N*′-tetramethylethylenediamine (TMEDA)). This was stopped after a defined time (CRP time) by adding the strong radical scavenger 2,2,6,6-tetramethylpiperidine 1-oxyl (TEMPO). Re-initiation was induced by UV-LED curing after casting the controlled radical polymerized PBM (CRP-PBM) obtained together with the photo-initiator 1-hydroxy-cyclohexyl-phenyl-ketone (Irgacure 184) on the commercial membrane. The photo-initiator was activated by UV-LED light at a wavelength of 365 ± 5 nm (*Opsystec Dr. Gröbel,* Ettlingen, Germany) [6]. High irradiation intensities enabled a fast polymerization reaction thus reducing the membrane coating production time [13].

Polymerization is an exothermal process and heat is released during the chain growth propagation [13,14]. This effect was exploited to follow the PBM CRP process by temperature measurements using thermocouples type K (*OMEGA Engineering*, Deckenpfronn, Germany). Initial tests were performed using test flasks equipped with three and five thermocouples with different spatial arrangements (Figure 1). Thermal profiles during polymerization were investigated for a 10 and 1.5 g PBM solution. A magnetic stirrer enabled steady mixing at a variable mixing speed of 0, 1000 and 1500 rpm, respectively. The increased turbulence effects were studied along with the impact on the temperature propagation. Partially polymerized PBM spots and agglomerates impaired uniform distribution on the membrane surface and reduced reproducibility. Therefore, a homogeneously dispersed polymerized PBM was of utmost importance in achieving a consistent coating quality. Passive cooling was enabled via natural convection in ambient air (Figure 1, left), while active cooling was realized with forced convection, a water bath and an installed heat exchanger connected to a thermostat *FP35* (*JULABO*; Figure 1, right). Continuous stirring supported the water flow conditions to maintain constant temperature and a higher heat dissipation. Ambient temperature was 22 ± 1 °C, while the water bath temperature was held constant at 20 ± 0.5 °C. Inert conditions inside the test flask were achieved using a gas bottle containing pressurized nitrogen (N_2_). With a better understanding of polymerization propagation inside the air-cooled (RT) 10 g flask, final experiments were duplicated with 1.5 g PBM sample volumes to reduce material input.

### 2.5. Rheological Investigations

Rheological investigations were carried out with the rheometer HAAKE RheoStress 1 (*Thermo Fisher Scientific*, Waltham, MA, United States). Measurements were conducted with a 35 mm, one slope titanium (Ti) cone. A 0.25 mL liquid was filled into the gap between the cone and the measuring plate before conditioning to the desired temperature of 20 °C via a Peltier element. Incremental step tests for stationary value assessment followed with controlled shear rates of 0 to 7000 s^−1^. The total test period was 300 s composing of 12 steps. Reference values were taken for the unpolymerized PBM mixture and compared to the viscosity of the CRP-PBM across different controlled radical polymerization times Δt_i_.

### 2.6. Nuclear Magnetic Resonance Spectroscopy 

The ^13^C solid-state nuclear magnetic resonance (NMR) spectra were run with an *Avance 300* spectrometer (*Bruker*, Billerica, MA. United States) at 75.5 MHz using magic angle spinning (MAS) with 4 mm rotor and a spinning rate of 10 kHz [15]. The cross-polarization (CP) ^13^C MAS-NMR spectra exploited the transfer of polarization under Hartmann–Hahn conditions obtained by the ramped amplitude variation of spin locking field (RAMP; [16]) and two-pulse phase-modulation (TPPM) decoupling [17]. The contact time of 1 ms and a recycling delay of 4 s were used and 1000 scans were acquired on average. In comparison, spectra were also achieved without CP with a HPDEC (high power proton decoupling) sequence of 2048 scans using a recycle delay of 2 s [18]. Due to the strong superposition of the resonances, the analysis of the spectra was focused to quantify only the intensity of resonances univocally associated to a specific molecular entity.

### 2.7. Membrane Coating Process

The membrane coating process and UV-LED curing of the controlled radical polymerized PBM (CRP-PBM) layer was carried out in a glovebox inflated with nitrogen (O_2_ < 1 w%) to avoid termination reactions caused by oxygen radicals during the polymerization process [19]. Process sequences were conducted in an automated homemade membrane coating machine (Figure 2) using a spiral casting knife with a 4 ± 2 µm wet layer thickness (*AB305X, TQC*) and a 6 cm·s^−1^ coating speed. UV-LED curing was carried out at 24 ± 1 °C and a 60 s UV-LED irradiation time at a 300 mW·cm^−2^ irradiation intensity (*Opsytec Dr. Gröbel*, Ettlingen, Germany). The UV-LED curing of the CRP-PBM was realized with Irgacure 184 and small quantities of APS as a TEMPO radical catcher.

### 2.8. Model Foulant Tests

Membrane performance tests using humic acid (HA) powder, purchased from *Sigma Aldrich* (St. Louis, MO, United States), as a model foulant were conducted within a membrane cross-flow unit (*SIMA-tec*). The cross-flow was set to 28 L·h^−1^ corresponding to a cross-flow-velocity of 0.1 m∙s^−1^. Prior to the permeability tests, the membrane samples of 0.0085 m^2^ were rinsed three times for one hour in deionized water (<20 µS·cm^−1^) to remove production-related glycerin. The test protocol at a 0.5 bar constant transmembrane pressure (TMP) was set as follows: (1) Pore acclimation with DI-water; (2) model foulant tests with a 100 mg·L^−1^ HA solution (pH 9); (3) calculation of the permeability using the last 20 volume flow records after reaching steady state conditions. The humic acid solution was prepared using ultrapure water (< 10 µS·cm^−1^) without any pH adjustment. A pH of 9 formed spontaneously. In this condition, humic acid is equally suspended and dissolved, causing the desired fouling effects. The sampling rate was set to 30 s^−1^ and the average value calculated. The permeability test for each CRP-PBM coating (Δt_i_ = 7, 8, 9 and 10 min) was duplicated. The system was flushed three times with DI-water before disassembly. A visual assessment of the fouling mitigation was carried out based on the brownish colored surface originating from the formation of a HA cake layer.

Constant pressure tests applied to membranes of different texture (pore structure) causes deviations in fouling propensity, as the flux rate is lower for more dense coating layers. Porous membranes with a higher flux commonly provoke fouling phenomena [20]. Therefore, the constant TMP operation only gives an indication of the structural layer density.

## 3. Results and Discussion

### 3.1. Pore Intrusion Identification in PBM-coated Membranes after UV-LED Polymerization

As stated in the introduction, PBM polymerization induced by UV-LED light led to a lower filtration capacity of the PBM-coated membrane in model foulant tests compared to redox-polymerized PBM-coated membranes [7]. The two polymerization attempts differ with regard to the process sequence. Specifically, the redox-polymerized-PBM was spread onto the membrane surface after 5 min of the polymerization initiation. Hence, already formed long-chain polymers increased the viscosity which changed the membrane wetting properties and thus the coating film characteristic. Additionally, according to Equation (1), a higher liquid viscosity reduces the capillary of a liquid–solid interface at a given pore size. In contrast, the UV-LED-PBM was spread onto the membrane prior to the polymerization initiation (start of UV-LED exposure) thus being more liquid in the casting process. Comparable performance for the UV-LED-polymerized-PBM should be achieved with an appropriate adjusted viscosity.

To verify that this was actually due to a higher intrusion into the membrane pores for UV-LED-polymerized PBM, we performed pore intrusion quantification experiments by visualizing fluorescent nanoparticles incorporated into the membrane. The manufacturer specifications give a nominal pore diameter of 35 nm and a maximum possible 0.1 µm pore size (*Martin Systems*). The total membrane thickness, as delivered, is indicated as 230 ± 20 µm (*Martin Systems*).

Figure 3 shows the cross-section of the UV-LED-polymerized PBM-coated membrane. Since the cast wet layer thicknesses were 4 ± 2 µm, the potential pore intrusion was calculated by subtracting the coated layer fraction from the total intrusion depth. This resulted in a maximum 60–80 µm intrusion depth along the cutting edge (Figure 3, green layer). The minimum pore intrusion was determined at around 26 µm.

The calculated values are based only on the visual observations using the confocal light microscope. Therefore, the values are just a rough estimate and provide qualitative prove of the high level of pore intrusion of the liquid PBM prior to the casting process. As the PBM tended to penetrate into the porous PES structure, the UV-LED polymerization caused partial or complete pore blockage. Aside from the increased membrane resistance through a higher membrane thickness, the decreased nominal pore diameter elevated the total membrane resistance substantially, leading to a lower permeability.

These observations prompted us to establish an experimental polymerization protocol that could suitably modulate the PBM viscosity prior to membrane casting and UV-LED curing in order to reduce the capillarity between liquid and solid porous membranes.

### 3.2. Controlled Radical Polymerization (CRP) of PBM

The basic premise was to verify the possibility of controlling the PBM viscosity by subjecting the PBM to a controlled radical polymerization stage (CRP) before it was cast on the surface of a commercial membrane and subjected to curing by UV-LED.

The goal was to develop a reproducible protocol for stopping the PBM polymerization within a defined time window prior to membrane coating. In this way the viscosity could be suitably increased, in order to reduce membrane pore intrusion during the coating stage. The method of adding TEMPO in order to retard and ultimately stop polymerization was adapted from Platkowski and Reichert [8]. The polymerization was stopped almost instantaneously after the addition of the very finely ground inhibitor TEMPO to the test tube. The inhibitor amount was gradually reduced in order to find the lowest possible amount required to stop the reaction while still allowing for an effective re-initiation by UV-LED curing with a reasonable amount of initiator. Since the initiator and the inhibitor were mutually influential to the polymerization reaction, it was vital to optimize the ratio of these two materials.

The application time of the inhibitor dosage to the PBM solution was defined as the CRP time Δt_i_ and varied between 7, 8, 9 and 10 min, relative to the initial starting value after the first redox initiation. Adding TEMPO too early caused an interruption leading to insufficiently low viscosities, whereas a late dosage had no effect on the self-accelerating polymerization. Successful reaction termination allowed a controlled radical polymerized PBM (CRP-PBM) to be obtained.

Re-initiation was completed first with APS and TMEDA to determine the feasibility of re-starting the chemical reaction. Consequently, the interim periods between stopping and re-starting were kept short and the material input was reduced successively. Re-initiation was then carried out after casting the produced CRP-PBM on the commercial membrane surface, by using the photo-initiator Irgacure 184 and UV-LED light for activation. Small amounts of APS could help eliminate excess TEMPO radicals and thus improve the curing efficacy.

The self-accelerating nature of exothermal polymerization reactions is well known [13]. Figure 4a–c shows the exothermal temperature profiles recorded in a flask containing 10 g PBM. The data highlights the influence of turbulences inside the test tube at passive (ambient) cooling for a stagnant (0 rpm (a)) and stirred sample at 1000 (b) and 1500 rpm (c) mixing speed.

Much higher temperature deviations in the space (yellow area) for the stagnant PBM solution (max. ± 14.5 K) indicated a highly inhomogeneous polymerization, whereas the temperature distribution was reduced at 1000 rpm (max. ± 6.5 K).

The wider temperature distribution deviation band for less turbulent flow conditions could be explained by the self-accelerating chemical reaction enhanced by the Trommsdorff–Norrish effect [21]. During polymerization, the heat transfer coefficient for the solid PBM phase dropped compared to the liquid phase, which reduced the heat dissipation.

Moreover, the chances of processes being terminated by recombination became lower due to decreased movement ability of the growing polymers with increasing viscosity. Furthermore, higher temperatures accelerated the radical decay and the heat dissipation was reduced by an increased viscosity with a lower heat conduction coefficient [21]. These effects caused an inhomogeneous polymerization with higher spatial temperature deviations (Figure 4, orange deviation band) and partial polymer agglomerates for low turbulences. On the contrary, Figure 4 ((d), (e) and (f)) depicts the polymerization inside a 1.5 g PBM test tube. The significant lower peak temperatures and the small deviation band were the result of the lower PBM mass. The total reaction enthalpy was much lower and the higher surface-to-volume ratio of the test tube substantially reduced both the Trommsdorff–Norrish effect and temperature increase due to improved heat dissipation into the surroundings. Consequently, more even spatial temperature distributions (max. ± 1 K) were obtained, beneficial for the desired homogeneous polymerization and the result of an even coating layer using casting coating techniques.

Complete polymerization at high turbulences (1500 rpm) did not occur for either PBM masses. This behavior was demonstrated by the near constant temperature observed until minutes 56 and 60, respectively (Figure 4, bottom graphs (c) and (f)). After reducing the turbulence to 1000 rpm, the PBM polymerized spontaneously in line with the propagation of the other experiments. However, it was noticeable that the temperature development of the delayed polymerization showed lower values, for instance, given a 55 °C temperature compared with 69.6 °C of the 1000 rpm polymerization (10 g PBM, Figure 4b,c, respectively). It was assumed that the radical diffusion rate was strongly influenced by the high convection conditions, thus inhibiting molecule propagation and chain transfer reactions. Nevertheless, very few radical reactions occurred, reducing the energy release at 1000 rpm.

Spatial distribution of temperature for the 1.5 g water-cooled flask was within the measuring error range of ± 0.5 K as the heat dissipation was further heightened by the high heat convection of the water bath. This further reduced the impact of the Trommsdorff–Norrish effect during polymerization. Consequently, the peak temperatures were lower (Figure 4e). Therefore, subsequent polymerization experiments were conducted with the smaller PBM sample volume of 1.5 g with active cooling inside the water bath.

Feasibility studies on successful polymerization inhibition were performed. The finely ground TEMPO radical inhibitor was then added to the polymerizing PBM solution. Figure 5 shows two different experiments using different amounts of TEMPO. A 0.5 mg amount of TEMPO did not completely inhibit polymerization as polymerization restarted spontaneously at minute 27 (Figure 5, left). Doubling the TEMPO amount to 1 mg resulted in the polymerization stopping successfully (Figure 5, right), validated by a long-term dormancy. Initiator and inhibitor were mutually influential and minimizing the material input was of prime importance. A 0.9 mg amount of TEMPO was found to be the threshold for a successful, sustained polymerization inhibition.

Based on these findings, a PBM viscosity modification protocol could be established by PBM controlled radical polymerization. This protocol comprises the following sequence of actions, as depicted in Figure 6 (Temperature measurements are highlighted for the passive cooled PBM):(1)Polymerization initiation using 1.8 mg APS activated by 40.5 µL TMEDA in a water bath at a 20 °C constant temperature(2)Polymerization inhibition with 0.9 mg of TEMPO-dosage administered after a defined time from initiation (controlled radical polymerization time, Δt_i_ = 7, 8, 9 and 10 min). This leads to controlled radical polymerized PBMs (CRP-PBMs) of different viscosities depending on the polymerization time Δt.

The CRP-PBM (1 g) thus obtained was then cast on the commercial membrane and cured by UV-LED using 75 mg Irgacure 184 as the photo-initiator, together with APS (5 mg), which was necessary to eliminate any excess of TEMPO radicals that could inhibit the UV-LED—induced curing reaction.

### 3.3. Rheological Examination

As previously discussed, the viscosity has a significant impact on the capillary and should thus affect the pore intrusion of the PBM into the membrane pores. The dynamic viscosity describes the internal resistance of fluids and gases to flow and shear conditions, depending on temperature and pressure [22]. The higher the viscosity, the higher the friction forces between the lattice molecules of individual molecular layers.

The potential effect of a higher viscosity of the controlled radical polymerized PBM (CRP-PBM) was observed by thickening the PBM with the water soluble, biodegradable polyethylene glycol (PEG). Visible examinations validated a lower intrusion level through the reduced transparency of the coated substrate. It is known that a higher viscosity reduces the capillarity between a liquid and a solid substrate using the universal Equation (1).
(1)$$h=\frac{2\cdot \;\sigma }{\rho \cdot g\cdot r}=\frac{2\;\cdot \sigma \cdot \nu }{\eta \;\cdot g\cdot r}$$

Equation (1) represents the capillarity in mm equal to the ratio of the fluid’s surface tension σ (mN·m^−1^) and the fluid density ρ in g·cm^−3^, the acceleration of gravity (m·s^−2^) and capillary radius *r* (mm). Since the density ρ is proportional to the dynamic viscosity (η~ρ) [23], lower capillarity can be achieved with a higher viscous fluid.

The results of rheological investigations are highlighted in Table 2 for the original unpolymerized PBM (Δt = 0) and CRP-PBMs at different polymerization times (Δt = 5.5, 7, 8 and 9 min). Frictional variance produced from the under-filling or over-filling of the gap between the cone and the measuring plate could have caused these deviations. The results clearly indicate a successful modification from lower (ca. 7 mPa·s, for unpolymerized PBM) to higher viscosity values (8.1–9, 6 mPa·s, for CRP-PBM) thanks to the controlled radical polymerization (CRP) protocol.

### 3.4. Nuclear Magnetic Resonance Spectroscopy

Figure 7 shows the typical ^13^C cross polarization magic angle spinning (CPMAS) spectra recorded for the materials obtained after casting PBMs on a commercial membrane followed by UV-LED curing or redox polymerization, namely: (a) CRP-PBM-9 (∆t = 9 min) (material obtained after CRP of PBM for 9 min followed by UV-LED curing with 5 w% Irgacure 184 as photo-initiator); (b) UV-LED-1.8 (material obtained after polymerization of the PBM with UV-LED using 1.8 w% Irgacure 184 as photo-initiator); (c) UV-LED-5 (material obtained after polymerization of the PBM with UV-LED using 5 w% Irgacure 184 as photo-initiator); and (d) REDOX-PBM (material obtained after redox polymerization of the PBM for 5.5 min).

In all samples, the two resonances at 8.5 ppm and between 22–29 ppm are uniquely attributed to polymerized AUTEAB [1]. Meanwhile, the signals around 18.1 ppm are associated with polymerized MMA (PMMA), HEMA (PHEMA) and EGDMA.

CPMAS spectra could be interpreted qualitatively given that the intensity of the resonance is a function of the strength of the dipolar coupling between observing carbons and neighboring protons.

Furthermore, dipolar couplings are modulated by local mobility; thus, the incorporated carbonyl groups (C=O) of the components are under-represented due to the absence of directly bonded protons [24]. Moreover, methylene and methyl groups show different cross-polarization dynamics. However, since all the spectra are recorded with the same parameters, variations in intensity of these groups give a qualitative idea of the variation in the composition of the copolymer [25,26].

The intensity ratios between the 8.5 and 22–29 ppm signals were compared across all samples (see Table 3, Integrals of CP spectra) and a significant variation is seen. This result is counterintuitive, since these signals all originate from polymerized AUTEAB. At 8.5 ppm the methyl groups resonate; 22–29 ppm all the methylene groups of the undecyl chain resonate except the last one bonded to oxygen, thus their relative intensity should not vary from one sample to the other. One possible explanation is that the distribution of stereo sequences of PMMA of PHEMA could vary within the different polymerization approaches, thereby modifying the line shape of the methyl group resonance from 16 to 22 ppm and partially overlapping with methylenes of the undecyl chain. As shown by Wilhelm et al. [27] in the case of pure PMMA, the broadening of ^13^C NMR peaks in the solid state can be directly assigned to the effects of conformational disorder within the amorphous PMMA polymer. A variation in microstructure (distribution of stereo sequences) can significantly change the line shape. However, this should be ruled out eventuality since considerable changes in the stereo control of the polymerization requires a large variation in polymerization temperatures, polarity of the solvent or ability of the catalyst to control the stereochemistry of the polymerization [28]. When comparing the polymerization conditions used for the different PBM samples, it is reasonable to assume that the obtained copolymers are atactic. Thus, the other only explanation for the variation of the internal ratio of the polymerized AUTEAB signal is that the different samples were characterized by a slightly different dynamic of the undecyl chain.

In a truly quantitative spectrum, the intensity ratio of methyls/methylenes for the undecyl chains of polymerized AUTEAB is 3:9. In fact, no sample shows this ratio, as methylene signals are always more intense than expected, because usually CH_2_ groups cross-polarize more efficiently than CH_3_. For the REDOX–BPM coated sample, the signals at 22–29 ppm are very intense. This is a possible indication of a very strong dipolar coupling typical of ordered (almost crystalline) systems (vide infra HPDEC spectra with a 20 s recycle delay). Although the 3:9 ratio was never observed because CPMAS is not quantitative, in the spectra of UV-LED-1.8 and UV-LED-5 the methylene signals are lower although very similar in the two samples. The spectrum of CRP-PBM-9 is analogous to those of the last two systems and presents an even lower methylene peak. Interestingly, the CRP-PBM-9 and UV-LED 5 samples distinctly showed detectable signals in the range of 120 to 140 ppm, due to the presence of Irgacure residues inside the polymers.

The HPDEC spectra were recorded with 20 s of recycle delay. Amorphous polymers with a low glass transition temperature, in general, can be analyzed more quantitatively with this technique [29]. This was also confirmed by the nearly constant 1:3 ratio between the methyl groups and the non-direct bonding to nitrogen or oxygen methylenes of the polymerized AUTEAB molecules (8.5 and 22–29 ppm, respectively) for all samples. The same trend was found for the acrylate signals at 18.1 and 177 ppm. Under stoichiometric conditions, the peak intensity at 177 ppm should be equal to the sum of 1/3 of the integral of signal at 8.5 ppm plus the integral of the signal at 18.1 ppm. The agreement in this case is acceptable. However, the ratio between acrylates (MMA, HEMA and EGDMA) and the polymerized AUTEAB is more adequately quantified by the direct ratio between the methyl signals at 8.5 and 18.1 ppm. The highest amounts were obtained by the REDOX-BPM sample, whereas UV-LED-5 showed the lowest values.

MAS-NMR data demonstrate that while in REDOX-BPM the AUTEAB chains are arranged in rigid structures, possibly ordered, this feature is less pronounced in the other polymerized PBMs. Considering that the feed composition is the same for all the samples, this altered behavior should be attributed to the different reactivity of the monomers under the diverse polymerization conditions. In fact, it seems that in the REDOX polymerization the AUTEAB monomers have a slightly higher preference to react with the growing chain if the reactive monomeric unit is an AUTEAB unit, since homosequences of AUTEAB can favor the formation of ordered domains.

### 3.5. Model Foulant Tests and the Relationship Between the Individual Parameters

Figure 8a illustrates the permeability of the membranes obtained after UV-LED curing of CRP-PBM‒cast membranes (deriving from CRP-PBMs obtained after different CRP times) and of the membranes obtained after casting with unpolymerized PBM followed by UV-LED polymerization (Δt_i_ = 0) and by redox polymerization for 5.5 min (REDOX-PBM). The viscosities of the uncast PBMs, as obtained by rheological measurements (see Section 3.3) are also shown.

As can be seen, for UV-LED cured CRP-PBM‒cast membranes, the permeability (orange dotted line) was decreased by increasing the dynamic viscosity of the uncast CRP-PBMs (brown dotted line), which, in turn, increased with the CRP time. Moreover, the permeability of the REDOX-PBM membrane was significantly higher with respect to all UV-LED polymerized coated membranes.

Since the casting coating procedure was the same for all PBMs (using the spiral-casting knife with a 4 µm wet coating thickness), these quite unexpected results are conceivably due to the modification of the polymer structure of the coating layers obtained upon polymerization under different conditions, as observed by solid state NMR measurements (Section 3.4).

Figure 8b shows the same membranes after the model foulant tests. The brownish color illustrates the fouling propensity for each coated membrane. As previously mentioned, a lighter color did not necessarily lead to a better fouling mitigation per se, as the membrane tests were conducted at constant TMP. Yet, the results are further evidence that the intervention of CRP followed by UV curing caused the porous coating layer to become a relatively dense structure. Accordingly, a greater change in CRP time Δt_i_ resulted in a denser layer for water filtration application. This in turn led to lower fouling propensity although the filtration capacity was reduced. With this learned knowledge, membrane coating layers with different filtration properties could be produced for a specific wastewater composition without changing the coating equipment or the coating material input.

All in all, the redox-polymerized PBM-coated membrane showed the highest permeability at 0.5 bar TMP (Figure 8a). As confirmed by NMR, the redox polymerization gave a different polymeric structure of polymerized PBM compared to the materials obtained by CRP followed by UV-LED curing, which was more favorable in terms of permeability for the filtration of humic acid solutions (100 mg·L^−1^, pH 9). On the other hand, the membrane coated with UV-LED polymerized PBM (Δt_i_ = 0) possessed a higher pore intrusion and thus the pore blockage was more extended with respect to the redox-polymerized PBM-coated membrane.

## 4. Conclusions

This study highlights a novel approach for changing the viscosity of a polymerizable bicontinuous microemulsion (PBM) via controlled radical polymerization (CRP). The study investigated the PBM polymerization using the redox initiator ammonium persulfate in combination with *N*,*N*,*N*,*N*′-tetramethylethylenediamine and the inhibitor 2,2,6,6-tetramethylpiperidinyloxyl (TEMPO). 

The influence of turbulence inside the test tube during PBM polymerization was identified. High turbulence reduced the spatial temperature distribution during the polymerization by enhanced heat dissipation into the surroundings. Lower PBM masses of 1.5 g minimized the prevailing Trommsdorff–Norrish effect of exothermal polymer reactions compared to the 10 g PBM mass. Active test tube cooling inside a water bath with forced convection further increased the heat release leading to a more homogeneous polymerization. PBM polymerization was successfully inhibited using the TEMPO radical inhibitor and could be re-initiated after casting the controlled radical polymerized PBM (CRP-PBM) thus obtained on a commercial membrane followed by UV-LED curing using the photo-initiator 1-hydroxy-cyclohexyl-phenyl-ketone (Irgacure 184). Since the initiator and the inhibitor were mutually influential, both material inputs were minimized to achieve the most effective CRP. Based on this understanding, a robust protocol for modifying the PBM viscosity was established. Prior to membrane coating, the dynamic viscosity of CRP-PBM with different CRP times was measured using a rheometer, and the results showed that the viscosity could be increased using the established protocol.

The performance of the final CRP-PBM coated membrane, after UV-LED curing, was examined in water permeability tests using the model foulant humic acid (HA). The water permeability dropped by more than 50% for the membrane obtained from the CRP-PBM that possessed a 26% higher viscosity compared to the membrane obtained with REDOX polymerized PBM. In general, the permeability was inversely proportional to the PBM viscosity (WP ~ η^−1^), due to different polymeric frameworks, as suggested by solid-state NMR. This phenomenon, however, also resulted in a higher fouling mitigation potential indicated by a thinner fouling cover layer of humic acid. Membranes coated with redox polymerized PBM also showed a higher permeability compared to membranes coated with UV-LED polymerized PBM, due to a different polymeric structure, as confirmed by solid-state NMR.

The results of this investigation may be useful to further help tailoring the density and porosity of coated membranes to a desired level for specific applications using different polymer coatings obtained under different and controlled polymerization conditions. Future research efforts will emphasise the impact of the CRP time on the pore size and finally the coating thickness.

## Figures and Tables

**Figure 1 membranes-10-00246-f001:**
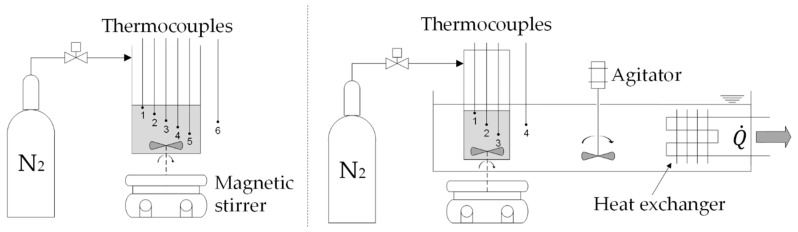
Setup for the polymerizable bicontinuous microemulsion (PBM) controlled radical polymerization investigation using temperature measurements.

**Figure 2 membranes-10-00246-f002:**
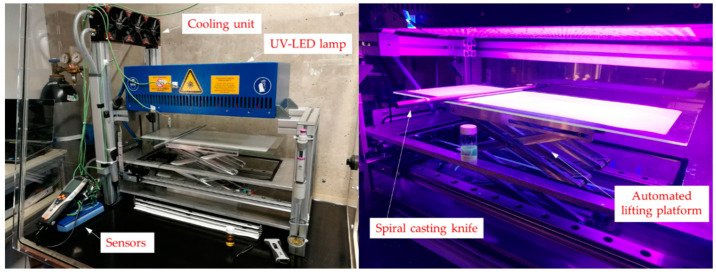
Automated membrane casting coating machine (right: irradiation).

**Figure 3 membranes-10-00246-f003:**
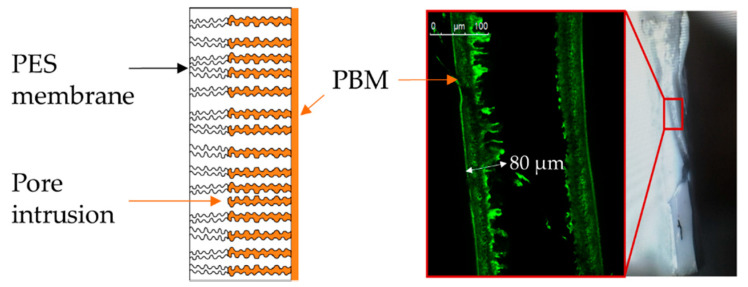
Confocal microscope tests revealing a high membrane pore intrusion level for the UV-LED-polymerized PBM-coated membrane, viewed in cross section.

**Figure 4 membranes-10-00246-f004:**
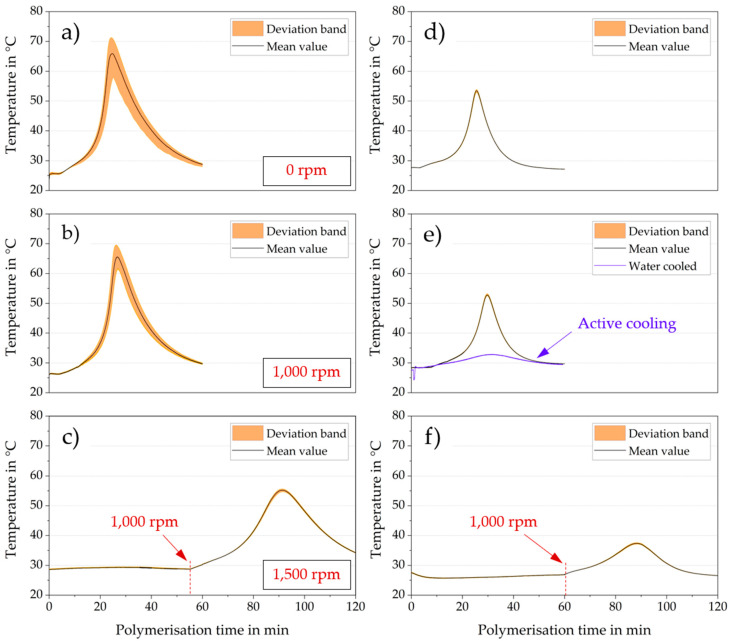
Temperature records inside a PBM solution subjected to redox polymerization (10 g for (**a**), (**b**), (**c**) and 1.5 g for (**d**), (**e**), (**f**) test volume).

**Figure 5 membranes-10-00246-f005:**
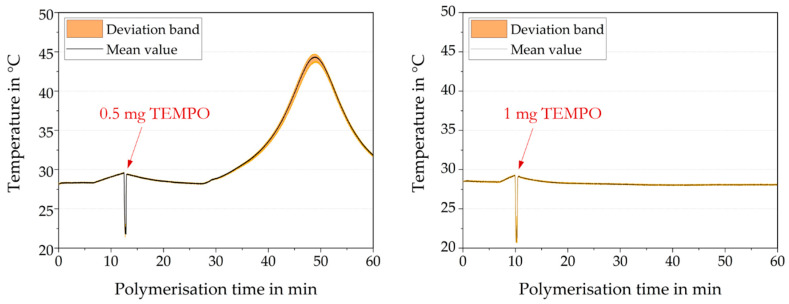
Temperature profiles of PBM redox polymerization using the TEMPO radical inhibitor at reported time.

**Figure 6 membranes-10-00246-f006:**
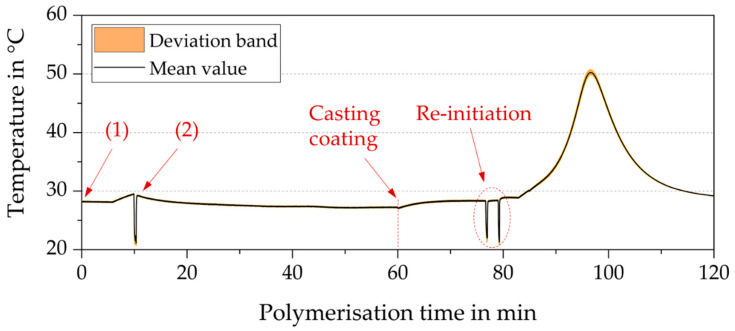
The temperature record of a 1 g PBM sample subjected to the controlled radical polymerization (CRP) protocol.

**Figure 7 membranes-10-00246-f007:**
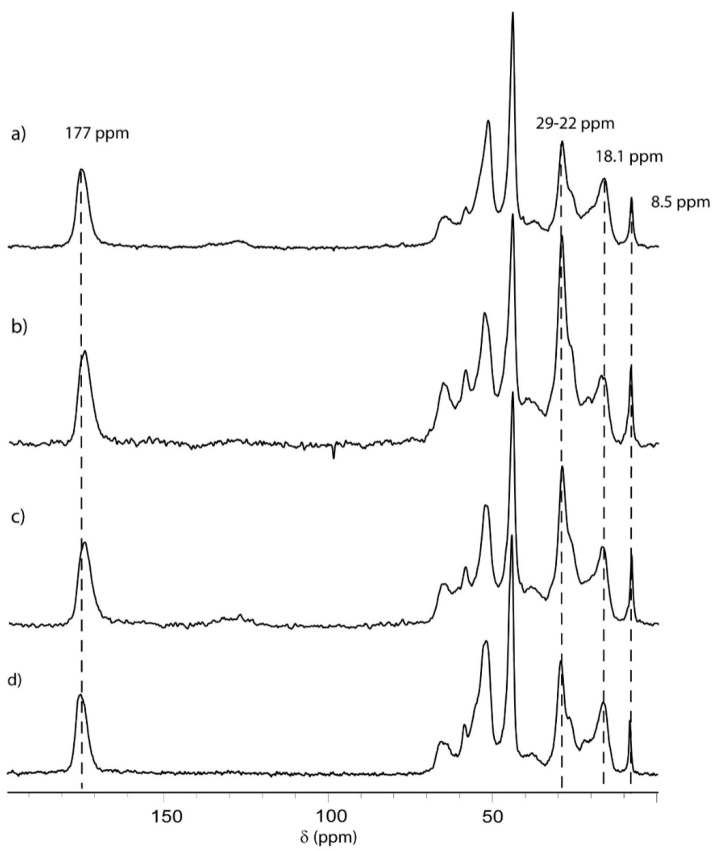
^13^C magic angle spinning nuclear magnetic resonance (MAS-NMR) of the analyzed samples: (**a**) CRP-PBM-9 (material obtained after CRP of PBM for 9 min followed by UV-LED curing with 5 w% Irgacure 184 as photo-initiator); (**b**) UV-LED-1.8 (material obtained after polymerization of the PBM with UV-LED using 1.8 w% Irgacure 184 as photo-initiator); (**c**) UV-LED-5 (material obtained after polymerization of the PBM with UV-LED using 5 w% Irgacure 184 as photo-initiator); and (**d**) REDOX-BPM (material obtained after redox polymerization of the PBM for 5.5 min)**.**

**Figure 8 membranes-10-00246-f008:**
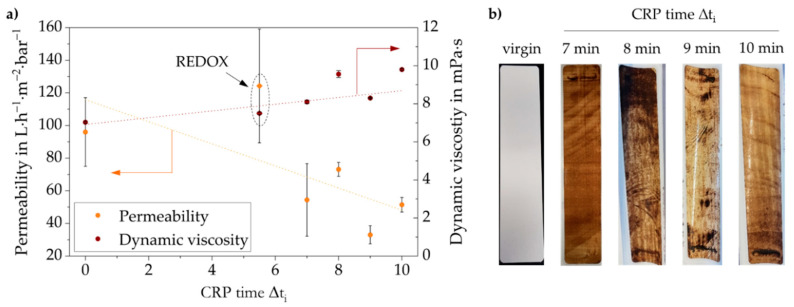
Membrane performance tests for different PBM viscosities and initiation temperatures.

**Table 1 membranes-10-00246-t001:** Chemicals used within this work.

Substance	Acronym	Function	Mass Fraction	Molecular Formula	Purity
Deionized water ^1^	DI water	Water phase	41 w%	H_2_O	<1.6 µS∙cm^−1^
Methylmethacrylate ^1^	MMA	Oil phase	21 w%	C_5_H_8_O_2_	99%
2-hydroxylethylmeth-acrylate ^1^	HEMA	Co-surfactant	10 w%	C_6_H_10_O_3_	≥99%
Ethylene glycol dimethacrylate ^1^	EGDMA	Cross-linker	3 w%	C_10_H_14_O_4_	98%
Acryloyloxyundecyl-triethyl ammonium bromide ^1^	AUTEAB	Main-surfactant	25 w%	C_20_H_40_BrNO_2_	>90%
1-Hydroxy-cyclohexyl-phenyl-ketone ^2^	Irgacure 184	Photo initiator	1.8, 2.3, 5 w% *	C_13_H_16_O_2_	99%
Ammonium persulfate ^3^	APS	REDOX initiator	0.12, 0.3 w% *	H_8_N_2_O_8_S_2_	≥98%
*N*,*N*,*N*′,*N*′-Tetramethylethylene-diamine ^3^	TMEDA	APS activator	2.7 w% *	C_6_H_16_N2	~99%
2,2,6,6-tetramethylpiperidine 1-oxyl ^4^	TEMPO	Inhibitor	0.1 w% *	C_9_H_18_NO	98%

* of total PBM mass.

**Table 2 membranes-10-00246-t002:** Viscosity measurements depending on CRP time.

Sample	Δt_i_	Viscosity	Deviation
	min	mPa·s	mPa·s
Unpolymerized PBM	0	7.03	± 0.05

REDOX-PBM	5.5	7.1	± 0.07

CRP-PBM	7	8.1	± 0.09
	8	9.6	± 0.18
	9	8.3	± 0.02

**Table 3 membranes-10-00246-t003:** Integrals of significant peaks in the ^13^C MAS-NMR spectra shown in Figure 7.

Sample Name	Integrals CP Signals			Integrals HPDEC Signals			
	8.5 (ppm)	18.1 (ppm)	22–29 (ppm)	8.5 (ppm)	18.1 (ppm)	22–29 (ppm)	177 (ppm)
CRP-PBM-9	1	3.7	7.9	1	0.86	3.25	1.19
UV-LED-1.8	1	2.7	9.4	1	-	-	-
UV-LED-5	1	3.3	9.2	1	0.55	2.89	0.82
REDOX-PBM	1	6.2	12.5	1	0.95	3.05	1.42

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
