# Peer review of "Viscosity Modification of Polymerizable Bicontinuous Microemulsion by Controlled Radical Polymerization for Membrane Coating Applications"

_membranes, 2020, doi:10.3390/membranes10090246_

Round 1
Reviewer 1 Report
The research focusses on controlling the viscosity of polymerizable bicontinuous microemulsions (PBM) before casting on commercial UF membranes. It demonstrates the important of coating solution viscosity on membrane preparation, especially in using PBM. The manuscript is acceptable after some minor revision. The following are my concerns:
- In introduction, the authors mentioned that “Polymerizable bicontinuous microemulsion (PBM) polymerized with redox initiation has demonstrated its potential as a low-fouling membrane coating for wastewater treatment application.” The authors could provide references for their claim.
- In introduction, the authors could provide more information about why PBM can provide low fouling tendency when coating onto membranes. For example, mentioning the materials used to fabricate PBM membrane that responsible for improving the antifouling property of the membrane.
- In Figure 2, the authors can include labels in the image of their automated membrane casting coating machine
- In section 3.1, the authors can emphasize that the viscosity of the coating solution affects the film forming into a substrate. Also, the pore size of the substrates affects the penetration of the casting solution. Hence, viscosity should be control. What is the pore size of commercial UF PES membrane used in this work?
- In Figure 8b, the author can also provide an image of the clean membrane (before fouling test), for comparison.
- How the CRP time affects the pore size, and final coating thickness?
Reviewer 2 Report
The reviewer has carefully read the manuscript titled of “Viscosity modification of polymerizable bicontinuous microemulsion by controlled radical polymerization for membrane coating applications” reported by Gukelberger et al. The authors report a method for controlling the viscosity of PBM before casting on commercial UF membranes for wastewater treatment applications. It is an interesting study. The reviewer thinks that this manuscript may be published after revision. Comments: 1. The “abstract” is confused and has no focus. It is difficult for readers to know what the authors want to emphasize. Please rewrite. 2. “shows the cross-section of the UV-LED polymerized PBM-coated membrane. Since the cast wet layer thicknesses was 4±2 µm, the potential pore intrusion was calculated by subtracting coated layer fraction from the total intrusion depth. This resulted in a maximum 60-80 µm intrusion depth along the cutting edge (Figure 3, green layer). The minimum pore intrusion was determined at around 26 µm.” Please provide the related data for supporting the results and describe the detail of obtaining values. 3. A schematic diagram of the key concept reported in this manuscript should be provide for better understanding.Author Response
Please see attachment.

Reviewer 3 Report
Review of the article entitled
Viscosity modification of polymerizable bicontinuous microemulsion by
controlled radical polymerization for membrane coating applications
Authors: Ephraim Gukelberger, Christian Hitzel, Raffaella Mancuso, Francesco Galiano, Mauro Daniel, Luigi Bruno, Roberto Simonutti, Bartolo Gabriele, Alberto Figoli, Jan Hoinkis
The subject of the manuscript is within the scope of Membranes journal. The paper is consistent and well organized as well as provides novel and interesting findings. The applied calculations and explanations mostly support the Authors' conclusions. I recommend to publish the manuscript after minor revision. Detailed comments are below.
- I highly encourage the Authors to provide the ranges of thicknesses of fabricated membranes.
- Scanning electron microscope photographs of fabricated membranes would provide valuable information regarding membranes pore structure and their distribution. Did the Authors record the SEM photographs?
- Why did the Authors perform the experiments in pH 9 solely?
- Line 213 – some editor error is visible.
- The manuscript should be carefully checked in terms of missing distances and grammar errors.
6. Yellow colour in Fig. 4 d,e,f (and so on) is really poorly visible, hence I encourage the Authors to improve the figures quality or change the colour.

Round 2
Reviewer 2 Report
The authors have addressed the issues.